# Effect of Sarcopenia on Survival and Health-Related Quality of Life in Patients with Hepatocellular Carcinoma after Hepatectomy

**DOI:** 10.3390/cancers14246144

**Published:** 2022-12-13

**Authors:** Jiawei Hu, Jinhuan Yang, Haitao Yu, Zhiyuan Bo, Kaiwen Chen, Daojie Wang, Yitong Xie, Yi Wang, Gang Chen

**Affiliations:** 1Department of Hepatobiliary Surgery, The First Affiliated Hospital of Wenzhou Medical University, Wenzhou 325035, China; 2Department of Epidemiology and Biostatistics, School of Public Health and Management, Wenzhou Medical University, Chashan High Education Zone, Wenzhou 325035, China; 3Key Laboratory of Diagnosis and Treatment of Severe Hepato-Pancreatic Diseases of Zhejiang Province, The First Affiliated Hospital of Wenzhou Medical University, Wenzhou 325035, China

**Keywords:** sarcopenia, hepatocellular carcinoma, hepatectomy, quality of life

## Abstract

**Simple Summary:**

Although sarcopenia reduces the quality of life in the general population, there is a lack of prospective cohorts in HCC patients to explore this relationship. We performed a prospective study to compare postoperative 1-year mortality and health-related quality of life after HCC resection between patients with and without sarcopenia. Our analyses revealed that patients with sarcopenia had a greater disease burden, which affected short-term postoperative survival rates and quality of life. Our study is the first step in exploring the relationship between sarcopenia and postoperative lifestyle in patients with liver cancer, which will further promote the implementation of preoperative sarcopenia assessment in patients with hepatocellular carcinoma.

**Abstract:**

Background: Although sarcopenia has been reported as a negative prognostic factor in patients with hepatocellular carcinoma (HCC), the lack of studies with a prospective design utilizing comprehensive sarcopenia assessment with composite endpoints is an important gap in understanding the impact of sarcopenia in patients with HCC. The aim of this study was to investigate the relationship between sarcopenia and postoperative 1-year mortality and health-related quality of life (HRQOL) based on sarcopenia assessment. Methods: The study cohort, who received resection surgery for HCC between May 2020 and August 2021, was assessed for sarcopenia based on grip strength, the chair stand test, skeletal muscle mass, and gait speed. The primary outcome measures were 1-year mortality and HRQOL determined using the QLQ-C30 questionnaire. In addition, we collected hospital costs, postoperative hospital stays, complications, 30-day and 90-day mortality, and 90- and 180-day readmission rates. Univariate and multivariate linear regression analyses were conducted to examine factors associated with global health status. Results: A total of 153 eligible patients were included in the cohort. One-year mortality was higher in patients with sarcopenia than in those without sarcopenia (*p* = 0.043). There was a correlation between sarcopenia and the surgical approach to global health status (*p* = 0.025) and diarrhea (*p* = 0.003). Conclusions: Preoperative sarcopenia reduces postoperative survival and health-related quality of life in patients with HCC.

## 1. Introduction

Sarcopenia, which is characterized by a progressive and generalized loss of muscle mass and muscle function [1], affects more than 50 million people worldwide, and the number of affected individuals is expected to exceed 200 million in the next 40 years [2]. Sarcopenia has been shown to be associated with nutritional health and to result in severely reduced health-related quality of life (HRQOL) [1,3,4]. A growing body of research has revealed that preoperative sarcopenia is associated with worse survival in patients with cancer [5,6,7,8,9,10,11,12]. However, numerous gaps exist in our current understanding of the association between preoperative sarcopenia and postoperative HRQOL in patients with cancer. Particularly, the majority of studies on sarcopenia are restricted to retrospective analyses and use a single measure of muscle mass to diagnose sarcopenia, which might render the assessment results relatively one-sided. According to the updated consensus guidelines of the European Working Group on Sarcopenia in Older People (EWGSOP2), a comprehensive evaluation of muscle mass and muscle strength should be used to assess sarcopenia [13]. Therefore, prospective and refined cohort studies with comprehensive sarcopenia assessments are urgently needed to assess the relationship between preoperative sarcopenia and postoperative HRQOL in patients with cancer [4].

Liver cancer is the sixth most common cancer and the third leading cause of cancer-related deaths worldwide, with 905,677 new liver cancer cases and 830,180 new deaths reported in 2020 [14]. Since its introduction in 1991 [15], laparoscopic hepatectomy has been demonstrated to be non-inferior to open hepatectomy for survival and recurrence and to be associated with fewer postoperative complications and shorter postoperative hospital stays [16,17,18]. Additionally, a study has found that the postoperative quality of life was better in patients with hepatocellular carcinoma (HCC) undergoing laparoscopic surgery than in those undergoing open surgery [19]. Regarding the quality of life after surgery in patients with cancer, the connection between the surgical method and sarcopenia is not well understood. Although a small number of studies have suggested that laparoscopic surgery for colorectal cancer might reduce the impact of sarcopenia on the poor quality of life after colorectal surgery [20,21], no study to date has examined the association between sarcopenia and the surgical approach in patients with other cancers such as HCC.

By using comprehensive methods to evaluate sarcopenia, we conducted a prospective study to compare postoperative 1-year mortality and HRQOL after HCC resection between patients with and without sarcopenia and to investigate differences in short-term HRQOL after specific surgical approaches according to sarcopenia. The present study findings should aid clinicians in further understanding the impact of sarcopenia on patients with HCC and in developing appropriate treatment plans for those with sarcopenia.

## 2. Materials and Methods

### 2.1. Cohort and Clinical Data

This was a prospective cohort study including consecutive patients who underwent liver resection for HCC between May 2020 and August 2021 in the First Affiliated Hospital of Wenzhou Medical University in Zhejiang, China (ClinicalTrials.gov identifier: NCT05339919) (Figure 1). Informed consent was obtained from patients and the study was approved by the Ethics Committee in Clinical Research of the First Affiliated Hospital. According to the EWGSOP2 guidelines, comprehensive data were collected to evaluate sarcopenia before surgery, including the SARC-F (sluggishness, assistance in walking, getting up from a chair, climbing stairs, falls) questionnaire, a lifestyle questionnaire, muscle strength (grip strength and five-time chair stand tests), muscle mass (skeletal muscle index (SMI)), and physical fitness (gait speed). Target patients were screened for eligibility to determine the final study population. Patients with a pathological diagnosis of HCC were eligible. Patients with missing data on sarcopenia and those who were planned to receive additional treatments either preoperatively or postoperatively were excluded. The treatment of HCC patients followed the standard for diagnosis and treatment of primary liver cancer issued by the National Health Commission of the People’s Republic of China [22]. Surgical approaches were classified as either open or laparoscopic. Laparoscopic and open surgery were selected according to the Chinese expert consensus on laparoscopic hepatectomy for hepatocellular carcinoma (2020 edition) [23]. The indications for laparoscopic liver resection for HCC are as follows:(1) Laparoscopic hepatectomy should be preferred for patients with HCC whose tumor diameter is ≤5 cm and located in the peripheral liver segment (II, III, IV, V, VI segment); (2) Tumor diameter < 5 cm, located in a difficult site, tumor diameter of 5–10 cm and multiple liver cancers meeting the Milan criteria can be implemented in experienced medical centers; (3) For massive HCC with a tumor diameter > 10 cm, laparoscopic liver resection can be performed in experienced medical centers after strict selection. Personal information such as medical insurance, work, education, and annual income of visiting patients was collected upon their admission. Clinical and surgical data were extracted from inpatient medical records after discharge, and information related to death and HRQOL were collected using follow-up interviews.

### 2.2. Assessment of Sarcopenia

According to the EWGSOP2 guidelines, the patients’ muscle condition was assessed using grip strength, a five-time chair stand test, and SMI. Based on these indicators, patients were categorized into sarcopenic and non-sarcopenic cohorts according to the EWGSOP2 guidelines [13]. Probable sarcopenia was defined as low grip strength or prolonged five-time chair standing tests. Patients with an SMI below the cutoff value in the probable sarcopenia population were diagnosed with sarcopenia.

Muscle strength was assessed using grip strength and a five-time chair standing test. Grip strength was measured twice for each hand, and the average of four trials was used for analysis [13,24]. The grip strength cutoff was 27 kg for male patients and 16 kg for female patients. The five-time chair stand test was conducted to measure the time it took for the patient to stand up from the chair five times in a row as quickly as possible. Subjects had to fully stand up and sit down each time without pausing, keeping their arms crossed in front of their chests [13,25]. The cutoff for the five-time chair stand test was 15 s for both male and female patients. SMI was defined as the total area of all skeletal muscles at the level of the third lumbar vertebra (psoas major, erector spinae, quadratus lumborum, transverse abdominis, external oblique, and internal oblique) divided by the square of the height (Appendix A). The SMI in this study was derived from the analysis of axial computed tomography (CT) images of the third lumbar region by two experienced radiologists. Presently, there are no uniform SMI cutoff values to define sarcopenia. The cutoff values for SMI were 51.1 cm^2^/m^2^ for men and 37.2 cm^2^/m^2^ for women, which we reported in previously published studies [7].

Gait speed was measured by calculating the speed of the patient walking on an 8 m straight line at daily speed in the hospital corridor [13,26]. The SARC-F scale score ≥ 4 was defined as symptomatic [27]. Moderate-to-high intensity exercise over 1 h/week or light-intensity exercise over 4 h/week is considered physical activity [28]. Diets were divided into three categories: vegetarian (excluding meat, poultry, and fish in the diet), semi-vegetarian (excluding red meat from the diet), and meat-eater (including red meat in the diet) [29].

### 2.3. Outcome measures

HRQOL at one year after surgery was the primary outcome of our study. To this end, the European Organization for Research and Treatment of Cancer QLQ-C30 questionnaire was collected using telephone interviews. The QLQ-C30 questionnaire is a short measure that has been translated into several languages and is recommended for patient evaluation after surgery for cancer [30]. Each question is scored on a scale from 0 (worst possible health state) to 100 (best possible health state) [31]. In addition, our study included hospital costs, postoperative hospital stay, complications, 90-day and 180-day readmission rates, and 30-day, 90-day, and 1-year mortality. We classified complication indicators into no complications, minor complications, and major complications according to the Clavien Dindo classification. Complications of grades I-II were defined as minor complications, and complications of grades III and above were defined as major complications [32].

### 2.4. Statistical Analysis

The Shapiro–Wilk test was used to assess the distribution of continuous variables. Normally distributed continuous variables and standardized HRQOL scores were presented as the mean (SD) and compared by analysis of variance. Continuous variables with a non-normal distribution were presented as the median (interquartile range [IQR]) and compared by the Kruskal–Wallis test, where statistically significant variables were subjected to Bonferroni correction. Categorical data were compared using the χ^2^ or Fisher’s exact test. The baseline demographic characteristics and clinical outcomes were compared between the patients with sarcopenia and those without sarcopenia in the general population. The baseline demographic characteristics and clinical outcomes were also compared between the patients undergoing laparoscopic surgery and those undergoing open surgery. The effect of surgical modality on HRQOL was compared by incorporating confounding factors in linear regression models. In addition, subgroup analysis of patients classified according to sarcopenia was performed to compare the differences between the laparoscopic and laparotomy groups in subgroups, and the results were further analyzed by regression analysis after adjusting for confounding factors. We also analyzed whether there is an interaction between surgical modality and sarcopenia on postoperative health-related quality of life. Finally, univariate and multivariate linear regression analyses were conducted to explore factors affecting the global health status of patients with HCC 1 year after surgery. All statistical analyses were conducted using R, a statistical software. *p* values were two-sided, with a *p*-value of ≤0.05 indicating statistical significance.

## 3. Results

### 3.1. Sarcopenia

The study cohort consisted of 153 eligible patients with a median age of 60.00 [IQR, 51.00–66.00] years, 133 male patients [86.9%], and 45 (29.4%) patients with sarcopenia (Figure 2a,b). A significant difference was observed in body mass index, grip strength, the chair stand test, and SMI between patients with sarcopenia and non-sarcopenia (*p* < 0.05 for all) (Figure 2c,e,f,g). Conversely, no significant difference was observed in SARC-F score, sleep time, diet, physical activity, abdominal circumference, and gait speed (*p* > 0.05 for all) (Figure 2d,h) (Table 1). People with sarcopenia had higher hospital costs (*p* = 0.014), longer postoperative hospital stays (*p* = 0.013), and higher one-year mortality (*p* = 0.043) compared to people without sarcopenia. Regarding postoperative complications, 90- and 180-day readmission rates, and 30- and 90-day mortality, no statistically significant results were observed in the two groups. (Table 1). Data on HRQOL were not collected for 24 (15.7%) patients. Therefore, these patients were excluded from the short-term postoperative quality of life assessment, and the HRQOL survey analysis included the remaining 129 patients. Figure 1 shows the box plot of the data collected using the QLQ-C30 questionnaire. The analysis revealed that the HRQOL collection was not significantly different between the sarcopenia and non-sarcopenia groups (*p* = 0.093) (Table 1).

### 3.2. The Surgical Approach

Surgery was an important factor affecting the short-term quality of life of patients after surgery (Appendix A). Among the 129 patients included in the analysis, the patients undergoing laparoscopy and open surgery differed in tumor TNM stage (*p* = 0.03), tumor size (*p* = 0.035), and satellite stove (*p* = 0.043) (Appendix A). Considering their confounding effect, TNM stage, tumor size, and satellite foci were included in the multivariate regression analysis. The patients who underwent laparoscopic surgery exhibited better outcomes in emotional functioning (*p* = 0.001), social functioning (*p* = 0.003), fatigue (*p* = 0.010), dyspnea (0.014), diarrhea (*p* = 0.032), and financial difficulties (*p* < 0.001) in the postoperative period than those who underwent open surgery (Appendix A).

### 3.3. Subgroup Analysis

The subgroup analysis of patients categorized according to sarcopenia revealed that the surgical approach had a greater and broader impact on postoperative quality of life in patients with sarcopenia than in those without sarcopenia (Appendix A). In order to assess the bias of the surgical approach in the sarcopenia population, we conducted a stratified analysis that divided sarcopenia patients into laparoscopic and open surgery subgroups to compare the baseline information. The result indicated that in the sarcopenia group, we found no statistically significant difference in age, sex, cirrhosis, child-cough grade, TNM stage, BCLC stage, tumor size, tumor location, tumor differentiation, satellite lesions, microvascular invasion, ASA grade, blood loss, albumin, and AFP between laparoscopic versus open surgery (Appendix A). It means that in the sarcopenia population in this study, there were no significant differences between laparoscopic and open patients in basic characteristics, liver function, tumor status, surgical procedure, and other nutritional status. The patients with sarcopenia who underwent laparoscopic surgery exhibited better outcomes in emotional functioning (*p* = 0.032), social functioning (*p* = 0.045), fatigue (*p* = 0.012), pain (*p* = 0.041), dyspnea (*p* = 0.022), diarrhea (*p* = 0.01), financial difficulties (*p* = 0.019), and global health status (*p* = 0.006) than those who underwent open surgery (Appendix A). In the non-sarcopenia group, TNM stage, tumor size, and satellite stove were included in the regression analysis to eliminate confounding effects on the surgical approach (Appendix A). Although there were differences in emotional functioning (*p* = 0.010), social functioning (*p* = 0.023), and financial difficulties (*p* = 0.006) between the laparoscopic and open surgery groups, fatigue (*p* = 0.136), diarrhea (*p* = 0.405), dyspnea (*p* = 0.174), pain (*p* = 0.265), and global health status (*p* = 0.680) did not differ significantly between the two groups (Appendix A).

### 3.4. Interaction of Sarcopenia with the Surgical Approach

After adjusting for confounders, the linear regression analysis evaluating the dimensions of the QLQ-C30 questionnaire revealed the presence of an interaction between sarcopenia and the surgical approach on global health status (*p* = 0.025) and diarrhea (*p* = 0.003) (Table 2). The multivariate linear regression on global health status, an indicator of overall HRQOL, revealed that serum albumin (<40 g/L: β = −13.29, 95% CI = −21.78, −4.79, *p* = 0.002) and the interaction between sarcopenia and the surgical approach (Sarcopenia laparoscopic surgery Group: β = 26.68, 95% CI = 9.96–43.39, *p* = 0.002; Non-sarcopenic open surgery group: β = 16.88, 95% = 2.31- 31.45, *p* = 0.024; non-sarcopenic laparoscopic surgery group: β = 20.97, 95% CI = 6.36–35.59, *p* = 0.005) were independent predictors of global health status (Table 3).

## 4. Discussion

In the present prospective study, which determined sarcopenia based on preoperative grip strength, the five-time chair stand test, and SMI according to the EWGSOP2 guidelines, we explored the relationship of sarcopenia with postoperative survival rate and HRQOL in patients with HCC. Our analyses revealed that patients with sarcopenia had a greater disease burden, which affected short-term postoperative survival rates and quality of life.

In our study, patients with sarcopenia had longer postoperative hospital stays and higher hospital costs compared with patients without sarcopenia. This revealed that patients with sarcopenia and HCC had weak perioperative recovery ability and lower perioperative quality of life. Previously, we reported that patients with sarcopenia had a higher risk of major complications [7]. Although the complication rate was not statistically significant in this study, we could find a trend towards higher rates of complications, including minor and major complications, in patients with sarcopenia. In addition, Giovanni Marasco et al. reported similar results that sarcopenia was associated with higher major complications (Clavien-Dindo grades III or IV) after resection of primary hepatocellular carcinoma in patients with compensated advanced chronic liver disease and portal hypertension [33].

We found a significant difference in 1-year mortality between patients with and without sarcopenia. De Buyser reported that sarcopenia was a predictor of mortality in older males [34]. As a new nutritional indicator, sarcopenia has attracted increasing attention but is not yet a routine assessment. Despite the use of different definitions of sarcopenia, sarcopenia has been reported to be associated with poor prognosis in various studies of HCC treatment, including liver resection, radiofrequency ablation, liver transplantation, and sorafenib systemic therapy [8,9,10,11,12]. However, the prognostic role of sarcopenia in other systemic treatments for advanced HCC, such as the approved regorafenib, remains to be further investigated [35]. Considering its impact on the prognosis of patients with HCC, preoperative assessment of sarcopenia should be promoted, and the nutritional status of patients before surgery should be improved [6,34,36,37].

Studies have demonstrated the association of HRQOL with prognosis in patients with HCC [38,39,40,41]. An increasing number of studies on liver tumors use patient-reported quality of life as a prognostic factor and clinical endpoint. This composite endpoint, which combines tumor-related and patient-centric experiences, will facilitate future clinical research and bring greater clinical benefit [42,43,44]. The surgical approach was a significant factor affecting the short-term quality of life after surgery in patients with HCC. Specifically, we found significant differences in postoperative emotional functioning, social functioning, fatigue, dyspnea, diarrhea, and financial difficulties between patients undergoing laparoscopic surgery and those undergoing open surgery.

Compared with patients without sarcopenia, the short-term quality of life after laparoscopic surgery was significantly improved in those with sarcopenia. Nutritional status is one of the most recognized determinants of surgical outcomes and is associated with poor postoperative outcomes [45]. Approximately 40% to 50% of patients undergoing surgery have some degree of malnutrition [46]. As a treatment, HCC induces inflammatory responses and related neurohumoral regulation, leading to the upregulation of inflammatory cytokines such as interleukin (IL)-1, IL-2, and IL-6. Simultaneously, the levels of anti-inflammatory cytokines such as IL-4 and IL-10 are also increased to induce a stress response [45]. The activation of both the inflammatory and anti-inflammatory responses negatively impacts metabolism, bone density, muscle mass and strength, exercise tolerance, the circulation system, cognition, and emotion, all of which significantly increase surgical risk and worsen outcomes. Additionally, dietary restrictions and immobilization in surgical patients can also increase muscle loss and malnutrition [45]. Therefore, we recommend that clinicians should evaluate sarcopenia before surgery and carefully select appropriate surgical methods for the best outcome, especially in patients with sarcopenia.

Our linear regression analysis of the specific dimensions of the QLQ-C30 questionnaire revealed an interaction between sarcopenia, the surgical approach to global health status, and diarrhea. This interaction, which has not been previously reported in patients with HCC, has important implications for treatment decision-making for patients with HCC and sarcopenia. The global health status is an officially recommended evaluation that examines the overall health status of patients [31]. In the present study, the univariate and multivariate linear regression analyses on the global health status revealed that the albumin level and the interaction between sarcopenia and the surgical approach were independent contributors to the global health status. As the most important protein in human plasma, albumin contributes to the maintenance of nutrition and osmotic pressure and is a commonly used nutritional status indicator in patients. Zoraida Verde reported that lower albumin levels were associated with self-reported mobility impairment and issues in daily activities, consistent with the present study findings [47]. In addition, the authors proposed an association between low albumin levels and low muscle strength. It is reasonable to expect correlations among nutritional indicators. Malnutrition associated with low albumin levels results in the degradation of protein and reduced muscle mass or function, i.e., strength and performance. Furthermore, sarcopenia and low serum albumin levels synergistically increase the risk of disability in older adults [48]. It is reasonable to speculate that patients with sarcopenia experience different levels of nutritional loss and recovery under different surgical modalities. In addition, the side effects of postsurgical cancer treatment, recovery from surgery, and reduced HRQOL associated with emotional burden may further negatively impact participation in daily life and lead to physical inactivity, accelerating the exacerbation of sarcopenia, which is more severe in patients with open surgery [49]. Ultimately, sarcopenia and surgical modalities interact to affect short-term HRQOL.

Considering the popularity of imaging tools for HCC patients, CT was used in this study to calculate SMI. Magnetic resonance imaging (MRI) is also considered the gold standard for the non-invasive assessment of muscle mass [13]. In addition, compared with CT, MRI can display the structure of muscle tissue and has a higher detection rate for smaller hepatocellular carcinomas. By combining various new technologies and new contrast agents, MRI can further improve the accuracy of the detection of sarcopenia. Matteo Renzulli et al. developed a new diagnostic algorithm for hepatocellular carcinoma in patients with chronic liver disease using gadolinium-ethoxybenzyl-diethylenetriamine pentaacetic acid [50]. More new research combined with novel imaging tools is required to develop diagnostic algorithms for sarcopenia for eventual application in clinical practice.

The present study has several notable strengths compared with the other studies on sarcopenia in patients with cancer. First, this was a prospective study that utilized a comprehensive set of indicators for patient evaluation. Most recent investigations examining sarcopenia in cancer have used a retrospective study design [4,5]. In these studies, the indicators of sarcopenia were single, failing to comprehensively evaluate sarcopenia and ultimately affecting the quality of the evidence. Additionally, the present study included not only postoperative survival but also the short-term quality of life in the first year after surgery. This is the first prospective study investigating the quality of life of patients with HCC and sarcopenia.

We acknowledge that the present study has several limitations. First, the patients were not randomly matched. Differences in clinician understanding and preference for surgical modalities may explain differences in baseline tumor characteristics between patients undergoing laparoscopic versus open surgery. To address this issue, we included tumor-related factors in the multivariate regression analysis. Additionally, we selected the most commonly used HRQOL questionnaire, the QLQ-C30, in the present study. A study has confirmed that the HRQOL-18 questionnaire can reflect the quality of life in patients with HCC [51], and future studies should consider other widely used questionnaires, such as the EQ-5D-5L [52]. Given that liver cancer is the sixth most common malignant tumor globally [14], further research on sarcopenia and HCC should include multicenter studies with large sample sizes. Finally, the present study evaluated HRQOL in the short-term postoperative period. As a prospective cohort, we will continue examining the relationship between sarcopenia and quality of life after surgery for HCC.

## 5. Conclusions

In summary, this is the first prospective cohort study demonstrating the negative impact of sarcopenia on postoperative 1-year survival in patients undergoing resection for HCC. Compared with laparoscopic surgery, open surgery was associated with significantly reduced HRQOL one year after surgery. Notably, this negative effect was worse in patients with sarcopenia. We also found an interaction between the sarcopenia status and the surgical approach on the short-term postoperative HRQOL. Albumin levels and the interaction of sarcopenia and surgery were the independent factors influencing the short-term overall HRQOL after surgery.

## Figures and Tables

**Figure 1 cancers-14-06144-f001:**
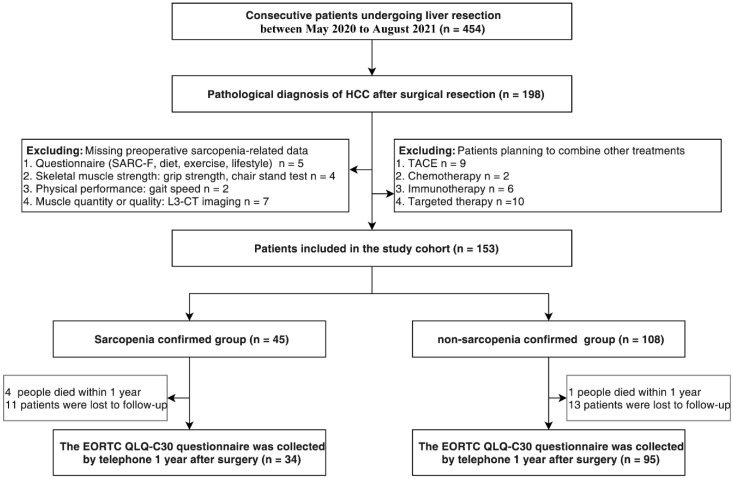
Screening of patients with HCC and sarcopenia study design.

**Figure 2 cancers-14-06144-f002:**
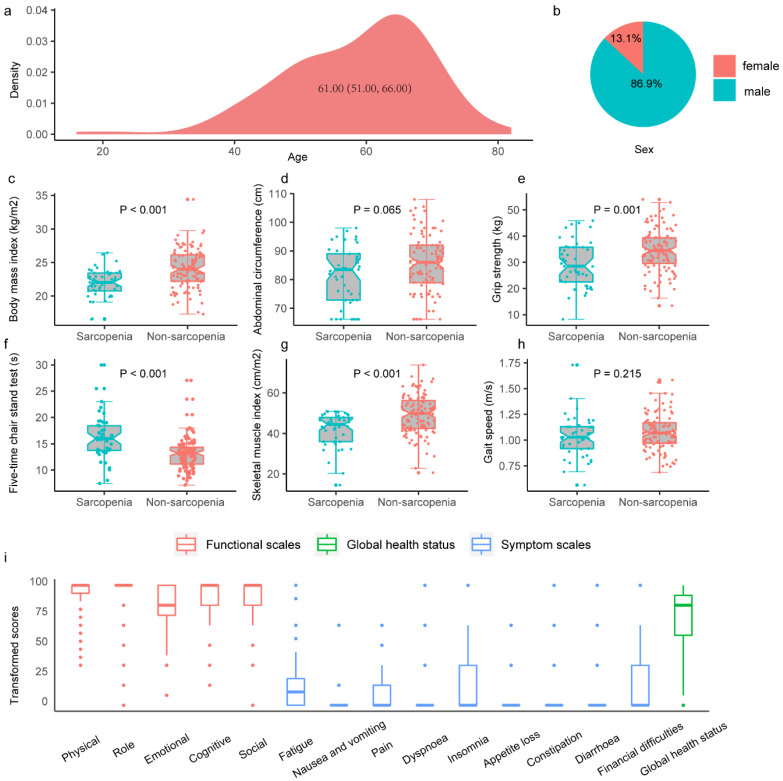
Informational description of the study cohort. Cohort age distribution (**a**) and sex distribution (**b**). (**c**–**h**) Description and comparison of sarcopenia-related indicators in people with and without sarcopenia: Body mass index (**c**), Abdominal circumference (**d**), Grip strength test (**e**), Five-time chair stand test (**f**), Skeletal muscle index (**g**), Gait speed (**h**). (**i**) Display of QLQ-C30 questionnaire data.

**Table 1 cancers-14-06144-t001:** Clinical characteristics and outcomes of the total population, patients with sarcopenia, and patients without sarcopenia.

Characteristics	Total Population N = 153	Patients with Sarcopenia N = 45	Patients without Sarcopenia N = 108	*p*-value
Age, n (%)				0.311
16–60	76 (49.7)	19 (42.2)	57 (52.8)	
61–82	77 (50.3)	26 (57.8)	51 (47.2)	
Sex, n (%)				0.210
Female	20 (13.1)	3 (6.7)	17 (15.7)	
Male	133 (86.9)	42 (93.3)	91 (84.3)	
Body mass index, n (%)				<0.001
<25 kg/m^2^	109 (71.2)	42 (93.3)	67 (62.0)	
≥25 kg/m^2^	44 (28.8)	3 (6.7)	41 (38.0)	
Abdominal circumference, cm	85.0 (76.0,91.0)	83.5 (72.8,89.0)	86.00 (78.9,92.0)	0.065
Grip strength test, kg	32.89 (8.60)	29.38 (8.83)	34.35 (8.11)	0.001
Chair stand test, s	13.6 (11.6, 15.9)	16.0 (13.3, 18.4)	13.2 (11.2, 14.3)	<0.001
Skeletal muscle index, cm/m^2^	47.04 (9.95)	41.84 (8.45)	49.20 (9.75)	<0.001
Gait speed, m/s	1.07 (0.96, 1.16)	1.03 (0.92, 1.13)	1.07 (0.97, 1.17)	0.215
High blood pressure, n (%)	50 (32.7)	14 (31.1)	36 (33.3)	0.938
Diabetes, n (%)	33 (21.6)	13 (28.9)	20 (18.5)	0.228
Heart disease, n (%)	7 (4.6)	3 (6.7)	4 (3.7)	0.708
Hepatitis B, n (%)	91 (59.5)	23 (51.1)	68 (63.0)	0.238
Physical activity, n (%)	99 (64.7)	29 (64.4)	70 (64.8)	1.000
SARC-F questionnaire, n (%)				0.154
≥4	151 (98.7)	43 (95.6)	108 (100)	
<4	2 (1.3)	2 (4.4)	0 (0.0)	
Sleep time, n (%)				0.737
≥8 h	35 (22.9)	9 (20.0)	26 (24.1)	
<8 h	118 (77.1)	36 (80.0)	82 (75.9)	
Diet, n (%)				0.527
Semi-vegetarian	58 (37.9)	16 (35.6)	42 (38.9)	
Meat-eater	63 (41.2)	17 (37.8)	46 (42.6)	
Vegetarian	32 (20.9)	12 (26.7)	20 (18.5)	
Smoking history, n (%)	97 (63.4)	32 (71.1)	65 (60.2)	0.274
Drinking history, n (%)	86 (56.2)	27 (60.0)	59 (54.6)	0.666
Medical insurance, n (%)	148 (96.7)	45 (100.0)	103 (95.4)	0.333
Employment, n (%)	65 (42.5)	14 (31.1)	51 (47.2)	0.097
Education, years				0.097
1–6	88 (57.5)	31 (68.9)	57 (52.8)	
6+	65 (42.5)	14 (31.1)	51 (47.2)	
Annual income, yuan				0.341
0–49,999	120 (78.4)	38 (84.4)	82 (75.9)	
50,000~	33 (21.6)	7 (15.6)	26 (24.1)	
TNM, n (%)				0.569
I	95 (62.1)	30 (66.7)	65 (60.2)	
II-IV	58 (37.9)	15 (33.3)	43 (39.8)	
BCLC, n (%)				0.729
0-A	104 (68)	32 (71.1)	72 (66.7)	
B-C	49 (32.0)	13 (28.9)	36 (33.3)	
Tumor size, n (%)				0.256
0.1 cm–5.0 cm	126 (82.4)	40 (88.9)	86 (79.6)	
≥5.1 cm	27 (17.6)	5 (11.1)	22 (20.4)	
Surgical approach, n (%)				1.000
Laparoscopy	77 (50.3)	23 (51.1)	54 (50.0)	
Open	76 (49.7)	22 (48.9)	54 (50.0)	
Transfuse blood, n (%)				0.446
No	77 (50.3)	20 (44.4)	57 (52.8)	
Yes	76 (49.7)	25 (55.6)	51 (47.2)	
Degree of differentiation, n (%)				0.286
Low	41 (26.8)	16 (35.6)	25 (23.1)	
Middle	92 (60.1)	24 (53.3)	68 (63.0)	
High	20 (13.1)	5 (11.1)	15 (13.9)	
AFP, n (%)				0.530
≤400 ng/mL	130 (85.0)	40 (88.9)	90 (83.3)	
>400 ng/mL	23 (15.0)	5 (11.1)	18 (16.7)	
Albumin, n (%)				0.185
≥40 g/L	79 (51.6)	19 (42.2)	60 (55.6)	
<40 g/L	74 (48.4)	26 (57.8)	48 (44.4)	
Costs, (yuan)	54189 (22598)	61132 (31947)	51297 (16645)	0.014
Postoperative stay, (day)	12.35 (5.45)	14.04 (7.51)	11.65 (4.17)	0.013
Complication, n (%)				0.162
No	74 (48.4)	17 (37.8)	57 (52.8)	
I-II	29 (19.0)	12 (26.7)	17 (15.7)	
III-IV	50 (32.7)	16 (35.6)	34 (31.5)	
90-day readmission, n (%)				0.449
No	118 (77.1)	37 (82.2)	81 (75.0)	
Yes	35 (22.9)	8 (17.8)	27 (25.0)	
180-day readmission, n (%)				0.339
No	109 (71.2)	35 (77.8)	74 (68.5)	
Yes	44 (28.8)	10 (22.2)	34 (31.5)	
Died within 30 days, n (%)				NA
No	153 (100.0)	45 (100.0)	108 (100.0)	
Yes	0 (0.0)	0 (0.0)	0 (0.0)	
Died within 90 days, n (%)				0.650
No	152 (100.0)	44 (97.8)	108 (100.0)	
Yes	1 (0.7)	1 (2.2)	0 (0.0)	
Died within 1 year, n (%)	5 (3.3)	4 (8.9)	1 (0.9)	0.043
HRQOL questionnaire, n (%)	129 (84.3)	34 (75.6)	95 (88.0)	0.093
Physical functioning	93.80 (13.54)	91.37 (15.44)	94.67 (12.78)	0.225
Role functioning	94.06 (16.77)	92.65 (17.01)	94.56 (16.74)	0.570
Emotional functioning	83.79 (16.38)	85.78 (15.29)	83.07 (16.78)	0.409
Cognitive functioning	90.83 (16.00)	93.14 (14.86)	90.00 (16.38)	0.328
Social functioning	89.53 (19.88)	84.31 (22.82)	91.40 (18.49)	0.074
Fatigue	15.68 (19.47)	17.97 (21.19)	14.85 (18.87)	0.425
Nausea and vomiting	0.90 (6.36)	0.00 (0.00)	1.23 (7.39)	0.336
Pain	8.01 (12.52)	4.41 (9.45)	9.30 (13.25)	0.050
Dyspnea	6.72 (16.86)	8.82 (22.19)	5.96 (14.57)	0.398
Insomnia	19.90 (31.05)	20.59 (33.85)	19.65 (30.17)	0.880
Appetite loss	3.88 (14.21)	3.92 (13.64)	3.86 (14.48)	0.983
Constipation	8.53 (20.95)	10.78 (24.23)	7.72 (18.48)	0.466
Diarrhea	3.62 (15.16)	7.84 (21.80)	2.11 (11.72)	0.058
Financial difficulties	25.06 (34.62)	25.49 (31.84)	24.91 (35.72)	0.934
Global health status	72.09 (25.76)	68.87 (28.15)	73.25 (24.91)	0.398

**Table 2 cancers-14-06144-t002:** Interaction between sarcopenia and the surgical approach on postoperative health-related quality of life.

Characteristics	β (95%CI)	*p*-Value
Physical functioning	−1.02 (−11.57, 9.54)	0.849
Role functioning	−3.59 (−17.06, 9.88)	0.599
Emotional functioning	−2.45 (−15.34, 10.43)	0.707
Cognitive functioning	0.85 (−11.98, 13.68)	0.896
Social functioning	−5.77 (−20.86, 9.32)	0.450
Fatigue	11.49 (−3.55, 26.53)	0.133
Nausea and vomiting	−1.21 (−6.11, 3.69)	0.625
Pain	3.10 (−6.60, 12.80)	0.528
Dyspnea	12.63 (−0.29, 25.55)	0.055
Insomnia	−2.76 (−28.22, 22.70)	0.830
Appetite loss	2.25 (−8.90, 13.41)	0.690
Constipation	−10.56 (−27.01, 5.90)	0.207
Diarrhea	17.82 (6.18, 29.47)	0.003
Financial difficulties	3.43 (−22.57, 29.44)	0.794
Global health status	−23.08 (−43.22, −2.94)	0.025

**Table 3 cancers-14-06144-t003:** Linear regression analysis of global health status in the patients with HCC.

Characteristics	Univariate Analysis *p*-Value	Multivariate Analysis	
		β (95%CI)	*p*-Value
Age			
16–60	Reference		
61–82	0.074		
Sex			
Female	Reference		
Male	0.888		
Physical activity			
No	Reference		
Yes	0.826		
Sleep time			
≥8 h	Reference		
<8 h	0.912		
Diet			
Semi-vegetarian	Reference		
Meat-eater	0.228		
Vegetarian	0.723		
Smoking history			
No	Reference		
Yes	0.410		
Drinking history			
No	Reference		
Yes	0.751		
Hypertension			
No	Reference		
Yes	0.221		
Diabetes			
No	Reference		
Yes	0.118		
Medical insurance			
No	Reference		
Yes	0.173		
Employment			
No	Reference		
Yes	0.203		
Education, years			
1–6	Reference		
6+	0.064		
Annual income, yuan			
0–49,999	Reference		
50,000~	0.248		
SARC-F			
≥4	Reference		
<4	0.536		
Sarcopenia			
No	Reference		
Yes	0.398		
Preoperative ascites			
No	Reference		
Yes	0.812		
Cirrhosis			
No	Reference		
Yes	0.580		
Child cough			
A	Reference		
B	0.921		
TNM			
I	Reference		
II-IV	0.034		
Tumor size			
0.1–5.0 cm	Reference		
≥5.1 cm	0.556		
ASA grade			
I–II	Reference		
III	0.065		
Blood loss			
≤400 mL	Reference		
>400 mL	0.017		
Transfuse blood			
No	Reference		
Yes	0.064		
BCLC			
0–A	Reference		
B–C	0.123		
AFP			
≤400 ng/mL	Reference		
>400 ng/mL	0.953		
Location			
Left liver	Reference		
Right liver	0.126		
Both sides	0.656		
Lesions			
Solitary	Reference		
Multiple	0.172		
Surgery			
Laparoscopy	Reference		
Open	0.033		
Degree of differentiation			
Low	Reference		
Middle	0.987		
High	0.132		
Satellite stove			
No	Reference		
Yes	0.302		
Vascular invasion			
No	Reference		
Yes	0.426		
Albumin			
≥40 g/L	Reference	Reference	
<40 g/L	0.003	−13.29 (−21.78, −4.79)	0.002
Interaction of sarcopenia and surgical approach			
Sarcopenia open surgery group	Reference	Reference	
Sarcopenia laparoscopic surgery group	0.003	26.68 (9.96, 43.39)	0.002
Non-sarcopenic open surgery group	0.023	16.88 (2.31, 31.45)	0.024
Non-sarcopenic laparoscopic surgery group	0.005	20.97 (6.36, 35.59)	0.005

Abbreviations: ASA, American Society of Anesthesiologists; AFP, alpha-fetoprotein.

## Data Availability

The data presented in this study are available upon request from the corresponding author.

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
