# Peer review of "Effect of Sarcopenia on Survival and Health-Related Quality of Life in Patients with Hepatocellular Carcinoma after Hepatectomy"

_cancers, 2022, doi:10.3390/cancers14246144_

Round 1

Reviewer 1 Report (Previous Reviewer 1)

Manuscript has improved with changes 

Good review 

This manuscript is a resubmission of an earlier submission. The following is a list of the peer review reports and author responses from that submission.

Round 1

Reviewer 1 Report

Simple summary: "Although sarcopenia has been shown to be significantly associated with healthy 15 quality of life" I do not agree with this sentence

Abstract: number of patients should be included in results not in methods

Methods: define the reasons to choose laparoscopic and open approach. Define Child/BCLC/ALBI or algorithm used to decide surgery as therapeutical approach. Why laparoscopy or open approach any hospital guidelines? Complications measured by Clavien Dindo? Mortality postoperative rate at 30 and 90 days? As you perfectly know complications, readmission are crucial for quality of life and have not been measured in your study This is my first major concern.

Results: As you say a bias in patients treated by lap or open approach is present. Perhaps patients with sarcopenia treated by lap approach are the best patients in sarcopenia group or better cases This my second major concern 

Author Response

Reviewer #1:

Simple summary

Comment: "Although sarcopenia has been shown to be significantly associated with healthy quality of life" I do not agree with this sentence

Response: Many thanks to you for taking out of your busy schedule to review our manuscript and put forward valuable suggestions for us. We agree with the comment and rewrote the sentence in the revised manuscript as the following” Although sarcopenia reduces quality of life in the general population”. Our statement is consistent with EWGSOP2 expression (Cruz-Jentoft, et al. Age Ageing. 2019 10.1093/ageing/afz046), based on the findings of a questionnaire study by Charlotte Beaudart et al (Beaudart, et al. J Cachexia Sarcopenia Muscle. 2017 10.1002/jcsm.12149).

(Line 15, page 1) Although sarcopenia reduces quality of life in the general population, there is a lack of prospective cohorts in HCC patients to explore this relationship.

Abstract

Comment: number of patients should be included in results not in methods

Response: We sincerely appreciate your nice suggestion. Considering your suggestion, we have revised the Methods and Results sections in the Abstract as follows:

(Lines 28-29, page 1) Methods: The study cohort, who received resection surgery for HCC between May 2020 and August 2021, were assessed for sarcopenia based on grip strength, chair stand test, skeletal muscle mass, and gait speed.

(Line 33, page 1) Results: A total of 153 eligible patients were identified for the cohort.

Methods:

Comment 1: define the reasons to choose laparoscopic and open approach. Define Child/BCLC/ALBI or algorithm used to decide surgery as therapeutical approach. Why laparoscopy or open approach any hospital guidelines?

Response 1: Thank you for taking the time to review our manuscript so carefully and for your valuable comments. The diagnosis and treatment of HCC patients followed the standard for diagnosis and treatment of primary liver cancer (2022 edition) issued by National Health Commission of the People’s Republic of China (General Office of National Health Commission. JOURNAL OF CLINICAL HEPATOLOGY. 2022 10.3969/j.issn.1001-5256.2022.02.009). Based on the patients' physical activity status, liver tumors and liver function, a staging scheme for liver cancer in China was established (China liver cancer staging, CNLC). Indications for liver resection include CNLC stage Ia, CNLC Ib, and CNLC IIa liver cancers with good liver reserve. For CNLC stage IIb, CNLC IIIa, and CNLC IIIb patients, Surgery is not the first choice for most patients, but may be considered in certain circumstances. According to the guidelines, laparoscopic liver resection has the same indications and contraindications as open surgery. In the actual diagnosis and treatment, the chief surgeon also considers the wishes of the patient and family members, and finally decides the surgical method. In this study, possible confounding factors have been incorporated into multivariate regression analysis to eliminate the confounding effects of tumor characteristics and socioeconomic and cultural background.

Comment 2: Complications measured by Clavien Dindo? Mortality postoperative rate at 30 and 90 days? As you perfectly know complications, readmission are crucial for quality of life and have not been measured in your study This is my first major concern.

Response 2: Thank you very much for your careful perusal of our article. Complications have been documented in patients in our cohort. According to the Clavien Dindo classification, grade 1-2 complications were defined as minor complications, including wound infection (bedside), nausea, vomiting, and elevated blood pressure; grade 3 or higher complications were defined as major complications, including postoperative thoracic complications Effusion (excluding reactive pleural effusion in patients with right hepatectomy), bile leakage, postoperative bleeding, liver failure, and death. We also collected length of postoperative hospital stay and hospital costs, as well as 90-day readmission. Compared with patients without sarcopenia, patients with sarcopenia had higher postoperative hospital complications, higher hospital costs, and longer postoperative hospital stays. There were no significant differences in 90-day readmission, 30-day mortality, and 90-day mortality. We have reported our findings in a previously published study (Yang, et al. J Cachexia Sarcopenia Muscle. 2022 10.1002/jcsm.13040). Given space constraints, we will present these published findings in the discussion. Thank you for your suggestion, we will add the information that needs to be explained as follows:

(Lines 226-230, pages 10-11) Previously, we reported that in patients with HCC, patients with sarcopenia had a higher risk of major complications, longer postoperative hospital stays, and higher hospital costs compared with patients without sarcopenia. This revealed that patients with sarcopenia and HCC had weak perioperative recovery ability and lower perioperative quality of life.

Results:

Comment: As you say a bias in patients treated by lap or open approach is present. Perhaps patients with sarcopenia treated by lap approach are the best patients in sarcopenia group or better cases This my second major concern

Response: Thank you very much for your willingness to spend a lot of precious time to review our manuscript. By subgroup analysis of sarcopenia patients, we found that there was no significant difference in baseline data between laparoscopic surgery patients and open surgery patients (all P>0.05), which means that there is no statistical bias. The results are presented in table S2.

Reviewer 2 Report

SPECIFIC COMMENTS

This manuscript is globally professionally written and concise although a few sections could be slightly improved to realize a more complete paper. There are minor typing errors along the manuscript that should be fixed. This paper perfectly reflects the authors guideline provided on the website of this prestigious journal.

Please note that it could be advisable to obtain a certified native speaker with proficiencies in the scientific-medical field to properly proofread this paper and to adopt a register adequate to this important journal.

ABSTRACT

The abstract is also reflecting the “Author Guidelines” reported in the journal. The main topics covered in this article are perfectly pre-viewed in the abstract and it encourages the reader to explore the entire paper. The abstract is clear and well-structured, and it reports only the main relevant information from the main text.

KEYWORDS

If possible, in order to enhance the finding process of this paper (and consequently the Journal) by readers and stakeholders, please try to check the selected keywords on MeSH Browser. For example, it could be possible to turn the term “Health-related quality of life” in “Quality of life” as indicated on MeSH Browser.

INTRODUCTION

In page 2 lines 44-45 authors stated that A “growing body of research has revealed that preoperative sarcopenia is associated with worse survival”. I agree with this sentence, but I also think that is important contextualize it and specify that other Authors also disclosed part of this argument, as demonstrated in recent papers such as [Cancers (Basel). 2022;14(8):1935. doi:10.3390/cancers14081935] // [J Gastroenterol. 2020;55(10):927-943. doi:10.1007/s00535-020-01711-w]. I think that authors could more widely disclose this argument and cite this important literature in this section or along discussion.

METHODS

In this paragraph the authors did not specify which radiological technique was used to assess the SMI. This is an essential information not to be missed in my opinion in this section that could also be disclosed in the discussion.

Authors described the others eligibility criteria, search strategy, study selection and other procedural patterns with great precision and without leaving any doubts on these arguments.

RESULTS

Results are quite clear, and easy to understand. All the significances are well-displayed although it would be more aesthetically pleasing to give greater prominence to the results with statistical significance by listing and analyzing them before results with non-statistical significance.

Moreover, my question for the authors is if they thought to complete a further subgroup analysis to investigate whether the impact of these results is also valid in patients undergoing first and second line systemic therapies, and not only in patients undergoing surgical therapy, as systemic therapies are used in many patients and have often proved effective and safe [Therap Adv Gastroenterol. 2021;14:17562848211016959. doi:10.1177/17562848211016959]. Authors can disclose this topic here or in the discussion section.

DISCUSSION

Also in this section, as mentioned before also could mention that their results are in line with those of extremely recent studies, also published in this important journal, focused on a remarkably similar topic as shown in [Cancers (Basel). 2022;14(8):1935. doi:10.3390/cancers14081935] // [J Gastroenterol. 2020;55(10):927-943. doi:10.1007/s00535-020-01711-w].

Moreover, authors never mentioned if, from a more radiological point of view, the valuation of sarcopenia via SMI, was better achieved using CT scans or MRI. This is living argument in so many fields of medicine, and thanks to the increasing use of MRI, I wondered if the authors had speculated on a possible use of specific MRI algorithm for the study of SMI as for example in HCC [J Gastroenterol Hepatol. 2016;31(1):69-80. doi:10.1111/jgh.13150].

TABLE AND FIGURES

The tables are simple and clear. The figures respect the quality standard and, in my opinion, they are very didactic and well-groomed.

FINAL COMMENT

In conclusion, I think this paper could certainly be considered as acceptable for publication after the completion of the suggested minor revisions.

Author Response

Reviewer #2:

SPECIFIC COMMENTS

Comment 1: This manuscript is globally professionally written and concise although a few sections could be slightly improved to realize a more complete paper. There are minor typing errors along the manuscript that should be fixed. This paper perfectly reflects the authors guideline provided on the website of this prestigious journal.

Response 1: We appreciate the reviewer’s positive evaluation of our work. We will carefully review the text of the article and correct any minor errors.

Comment 2: Please note that it could be advisable to obtain a certified native speaker with proficiencies in the scientific-medical field to properly proofread this paper and to adopt a register adequate to this important journal.

Response 2: We have commissioned certified native speakers in specialized fields to carefully review our manuscript. Once again, we asked them to help us review the manuscript before we submitted a revised version.

ABSTRACT

Comment: The abstract is also reflecting the “Author Guidelines” reported in the journal. The main topics covered in this article are perfectly pre-viewed in the abstract and it encourages the reader to explore the entire paper. The abstract is clear and well-structured, and it reports only the main relevant information from the main text.

Response: We appreciate the reviewer’s positive evaluation of our work.

KEYWORDS

Comment: If possible, in order to enhance the finding process of this paper (and consequently the Journal) by readers and stakeholders, please try to check the selected keywords on MeSH Browser. For example, it could be possible to turn the term “Health-related quality of life” in “Quality of life” as indicated on MeSH Browser.

Response: We sincerely appreciate your nice suggestion. We filtered through the MeSH browser and modified the keywords as follows:

(Lines 38, page 1) Keywords: Sarcopenia; Hepatocellular carcinoma; Hepatectomy; Quality of life

INTRODUCTION

Comment: In page 2 lines 44-45 authors stated that A “growing body of research has revealed that preoperative sarcopenia is associated with worse survival”. I agree with this sentence, but I also think that is important contextualize it and specify that other Authors also disclosed part of this argument, as demonstrated in recent papers such as [Cancers (Basel). 2022;14(8):1935. doi:10.3390/cancers14081935] // [J Gastroenterol. 2020;55(10):927-943. doi:10.1007/s00535-020-01711-w]. I think that authors could more widely disclose this argument and cite this important literature in this section or along discussion.

Response: Thank you very much for taking the time to review our manuscript and point out the shortcomings in our manuscript. As suggested by reviewer, we have added the suggested content to the discussion as follows:

(Lines 230-233, page 11) In addition, Giovanni Marasco et al reported similar results that sarcopenia was associated with higher major complications (Clavien-Dindo grade III or IV) after resection of primary hepatocellular carcinoma in patients with compensated advanced chronic liver disease and portal hypertension.

(Lines 237-240, page 11) Despite the use of different definitions of sarcopenia, sarcopenia has been reported to be associated with poor prognosis in various studies of treatment of HCC including liver resection, radiofrequency ablation, liver transplantation, and sorafenib systemic therapy.

METHODS

Comment 1: In this paragraph the authors did not specify which radiological technique was used to assess the SMI. This is an essential information not to be missed in my opinion in this section that could also be disclosed in the discussion.

Response1:Thank you for the suggestion. We have added the information required as follows:

(Lines 116-117, page 3) The SMI in this study was derived from the analysis of axial computed tomography (CT) images of the third lumbar region by two experienced radiologists.

Comment 2: Authors described the others eligibility criteria, search strategy, study selection and other procedural patterns with great precision and without leaving any doubts on these arguments.

Response2: Special thanks to you for your good comments.

RESULTS

Comment 1: Results are quite clear, and easy to understand. All the significances are well-displayed although it would be more aesthetically pleasing to give greater prominence to the results with statistical significance by listing and analyzing them before results with non-statistical significance.

Response1:  We appreciate the reviewer’s positive evaluation of our work.

Comment 2: Moreover, my question for the authors is if they thought to complete a further subgroup analysis to investigate whether the impact of these results is also valid in patients undergoing first and second line systemic therapies, and not only in patients undergoing surgical therapy, as systemic therapies are used in many patients and have often proved effective and safe [Therap Adv Gastroenterol. 2021;14:17562848211016959. doi:10.1177/17562848211016959]. Authors can disclose this topic here or in the discussion section.

Response2: Thank you very much for your careful perusal of our article. The aim of this study was to evaluate the association between sarcopenia and poor prognosis after hepatectomy in patients with hepatocellular carcinoma. We planned to construct a cohort of patients with sarcopenia and hepatocellular carcinoma who received systemic therapy, but the small sample size hindered our further studies. We will address this research gap as follows:

(Lines 240-242, page 11) However, the prognostic role of sarcopenia in other systemic treatments for advanced HCC, such as the approved regorafenib, remains to be further investigated.

DISCUSSION

Comment 1: Also in this section, as mentioned before also could mention that their results are in line with those of extremely recent studies, also published in this important journal, focused on a remarkably similar topic as shown in [Cancers (Basel). 2022;14(8):1935. doi:10.3390/cancers14081935] // [J Gastroenterol. 2020;55(10):927-943. doi:10.1007/s00535-020-01711-w].

Response1: Thank you very much for your precious time and energy to review our manuscript. As suggested by reviewer, we have added the suggested content to the discussion as follows:

(Lines 230-233, page 11) In addition, Giovanni Marasco et al reported similar results that sarcopenia was associated with higher major complications (Clavien-Dindo grade III or IV) after resection of primary hepatocellular carcinoma in patients with compensated advanced chronic liver disease and portal hypertension.

(Lines 237-240, page 11) Despite the use of different definitions of sarcopenia, sarcopenia has been reported to be associated with poor prognosis in various studies of treatment of HCC including liver resection, radiofrequency ablation, liver transplantation, and systemic therapy.

Comment 2:  Moreover, authors never mentioned if, from a more radiological point of view, the valuation of sarcopenia via SMI, was better achieved using CT scans or MRI. This is living argument in so many fields of medicine, and thanks to the increasing use of MRI, I wondered if the authors had speculated on a possible use of specific MRI algorithm for the study of SMI as for example in HCC [J Gastroenterol Hepatol. 2016;31(1):69-80. doi:10.1111/jgh.13150]

Response 2: Thank you for taking the time to review our manuscript so carefully and for your valuable comments. Undoubtedly, the use of MRI to calculate SMI is highly recommended. MRI and CT are considered the gold standard for noninvasive assessment of muscle mass (Cruz-Jentoft, et al. Age Ageing. 2019 10.1093/ageing/afz046). Considering that compared with CT, MRI can clearly display the structure of muscle tissue and has a higher detection rate for smaller hepatocellular carcinoma. In addition, liver MRI can also combine various new technologies and new contrast agents to further improve the accuracy of differential diagnosis of tissues. It is reasonable to speculate that SMI calculated in combination with MRI can improve the accuracy of the diagnosis of sarcopenia. We will elaborate on the discussion as follows:

(Lines 294-303, page 12) Considering the popularity of imaging tools for HCC patients, CT was used in this study to calculate SMI. Magnetic resonance imaging (MRI) is also considered the gold standard for non-invasive assessment of muscle mass. In addition, compared with CT, MRI can clearly display the structure of muscle tissue and has a higher detection rate for smaller hepatocellular carcinoma. By combining various new technologies and new contrast agents, MRI can further improve the accuracy of sarcopenia. As Matteo Renzulli et al. developed a new diagnostic algorithm for hepatocellular carcinoma in patients with chronic liver disease using gadolinium-ethoxybenzyl-diethylenetriamine pentaacetic acid. We call for more new research combined with novel imaging tools to develop diagnostic algorithms for sarcopenia for eventual application in clinical practice.

TABLE AND FIGURES

Comment : The tables are simple and clear. The figures respect the quality standard and, in my opinion, they are very didactic and well-groomed.

Response: We thank the reviewer for this positive appraisal.

FINAL COMMENT

Comment: In conclusion, I think this paper could certainly be considered as acceptable for publication after the completion of the suggested minor revisions.

Response: Many thanks to you for taking out of your busy schedule to review our manuscript and put forward valuable suggestions for us. Thanks for the positive reception of the manuscript and the positive comments.

Round 2

Reviewer 1 Report

The changes di not answere my suggestions and comments